# The Impact of New Surgical Techniques on Geographical Unwarranted Variation: The Case of Benign Hysterectomy

**DOI:** 10.3390/ijerph18136722

**Published:** 2021-06-22

**Authors:** Daniel Adrian Lungu, Elisa Foresi, Paolo Belardi, Sabina Nuti, Andrea Giannini, Tommaso Simoncini

**Affiliations:** 1Management and Healthcare Laboratory, Institute of Management and Department EMbeDS, Scuola Superiore Sant’Anna, 56127 Pisa, Italy; e.foresi@inrca.it (E.F.); paolo.belardi@santannapisa.it (P.B.); sabina.nuti@santannapisa.it (S.N.); 2Obstetrics and Gynecology, Department of Clinical and Experimental Medicine, University of Pisa, 56126 Pisa, Italy; andrea.giannini@unipi.it (A.G.); tommaso.simoncini@med.unipi.it (T.S.)

**Keywords:** elective surgery, waiting times, benign hysterectomy, unwarranted variation, treatment rate

## Abstract

Since the 1980s, the international literature has reported variations for healthcare services, especially for elective ones. Variations are positive if they reflect patient preferences, while if they do not, they are unwarranted, and thus avoidable. Benign hysterectomy is among the most frequent elective surgical procedures in developed countries, and, in recent years, it has been increasingly delivered through minimally invasive surgical techniques, namely laparoscopic or robotic. The question therefore arises over what the impact of these new surgical techniques on avoidable variation is. In this study we analyze the extent of unwarranted geographical variation of treatment rates and of the adoption of minimally invasive procedures for benign hysterectomy in an Italian regional healthcare system. We assess the impact of the surgical approach on the provision of benign hysterectomy, in terms of efficiency (by measuring the average length of stay) and efficacy (by measuring the post-operative complications). Geographical variation was observed among regional health districts for treatment rates and waiting times. At a provider level, we found differences for the minimally invasive approach. We found a positive and significant association between rates and the percentage of minimally invasive procedures. Providers that frequently adopt minimally invasive procedures have shorter average length of stay, and when they also perform open hysterectomies, fewer complications.

## 1. Introduction

The Italian National Health System (NHS) follows a Beveridge model and provides universal coverage. The regional healthcare system is organized into three (prior they were 12) Local Health Authorities (LHAs) with approximately 40 community hospitals, 26 health districts, and four Teaching Hospitals (THs).

Elective surgery volumes have increased significantly in recent years [1,2,3,4]. This increase has been caused by different factors, including the population’s higher life expectancy, advancement of technology, and the growing expectations and trust in the favorable outcomes of surgical treatments from patients [1,2,3,4,5]. Moreover, elective surgery represents the biggest portion of the national surgical Diagnosis-Related Groups (DRGs), accounting for approximately 2/3 of the total surgical activities [6].

Inter-regional variation persisted over time for most of the elective surgery services, as depicted by the National Outcome Evaluation Program [7] and by the Inter-Regional Performance Evaluation System data [5]. Tuscany is member of the Inter-Regional Performance Evaluation System, which currently includes 11 regional healthcare systems (in 2019) that measure and evaluate their multidimensional performance dimensions of public healthcare organizations, through systematic and publicly disclosed benchmarking [6,7]. Performance is measured through more than 300 key performance indicators. There are several indicators measuring surgical activity in terms of quality, appropriateness, and adoption of minimally-invasive techniques, and all indicators can be accessed on the online platform (in Italian—https://performance.santannapisa.it/ accessed on 1 June 2020) [5].

In economic terms, in 2018 the national health fund was equal to EUR 113.4 billion [8]. Out of it, approximately 45% was devoted to hospital care, and it is estimated that the expenditure for surgical care at the hospital level amounted to almost EUR 26 billion [9,10]. As elective procedures account for about 2/3 of the total surgical activity, the estimated value of elective surgery is equal to EUR 17.4 billion, thus representing a significant share of the overall healthcare expenditure.

Starting with the studies conducted by Jack Wennberg in the 1970s, investigating differences in care provision among small areas in Vermont [11], the topic of geographical variation has become one of the main research fields for healthcare management scholars. When unwarranted, variation conflicts with the three pillars of public healthcare systems: quality, equity and financial sustainability [12]. The main categories of geographical variation identified by Wennberg [13] and then integrated by Nuti et al. [14] are:
(1)Effective care: it includes all the services whose effectiveness has been demonstrated and for which benefits clearly outweigh risks.(2)Preference-sensitive care: it includes all those conditions where two or more clinically acceptable options exist, and choice should depend on patient preferences.(3)Supply-sensitive care: for these services, the amount of per capita resources allocated to a given population largely determines the frequency of use, although several studies have demonstrated that higher intensity of care is not associated with better health outcomes [15,16].(4)Poor integration along the care pathway: for some specific conditions, mainly chronic, geographical variation might depend upon the different extent of integration between the care settings, alongside the care pathway.

Additionally, the unwarranted geographical variation in the provision of elective care has been widely documented across and within countries [17,18]. Since elective surgical services are related to both preference-sensitive and supply-sensitive factors [14,19], some studies have suggested that policy-makers and healthcare managers should analyse unwarranted geographical variation in order to detect inappropriate care delivery, to improve quality of care and achieve equity of access to healthcare services [14,20,21,22].

Beside variation, in Beveridge-model public systems, policy-makers should also guarantee citizens equitable access to care, regardless of individual ability to pay or other characteristics such as income, region of residence, or other [14,23,24]. In a context of increased population age (and therefore needs), rapid technological advancements and slower economic growth, the Italian healthcare system faces the challenge of providing equitable, appropriate, and high-quality care while keeping an eye on the financial sustainability.

Regarding elective surgery services, policymakers and healthcare managers have to organize their delivery by finding the right balance between supply and demand. As decision makers’ are interested in obtaining consensus, their priority lies with finding solutions aimed at reducing long waiting lists, which have often entailed enhancing supply, by increasing capacity (e.g., operating room hours) [25,26]. While on one hand the issue of waiting times is considered relevant to obtain political consensus in the short term, on the other hand, existing literature has shown that these solutions have failed to reduce waiting times in the long run at system level [27,28,29].

Additionally, the increasing adoption of minimally-invasive surgical techniques (robotic and laparoscopic) has brought undoubtable clinical benefits such as reduced blood loss, less post-operative pain and better aesthetical outcomes [15], but at the same time has raised concerns over the cost-effectiveness and the contribution of such techniques to the widely discussed matter of clinical practice variation introduced above. Beside the benefits mentioned above, minimally-invasive surgery has proved in multiple international studies to be associated with shorter average length of stay with respect to the open surgical approach [30,31,32,33]. This is an important element to be kept in mind in relationship to the effort that policymakers are putting in place to shorten waiting lists, as a greater adoption of minimally invasive procedures might also contribute to a quicker turnaround for care providers.

Several studies have highlighted that there has been a rising adoption of minimally-invasive techniques for many surgical procedures [34,35,36]. The choice of hysterectomy as an index surgery is relevant, since it represents the most frequent gynecological surgical procedure in women and it is a highly standardized procedure, where the indications and outcomes are easy to be compared.

Through analyzing the provision of a specific elective surgical procedure in an Italian regional healthcare system, our study aims to investigate the impact of the surgical approach on the avoidable variation.

### Benign Hysterectomy: An Overview

Hysterectomy for benign diseases is among the most frequently performed elective surgical procedures in the developed Countries and it consists into the total or partial removal of the uterus [37]. Benign hysterectomies are by definition those procedures performed for benign gynecological conditions [38]. Of these, heavy menstrual bleeding is the most common, followed by uterine fibroids (leiomyoma) and pelvic organ prolapse, and, less commonly, endometriosis and adenomyosis [39,40].

The choice of the surgical approach depends upon a number of clinical and cultural factors, including clinical circumstances, surgeon’s technical expertise, and patients’ preferences; benign hysterectomy can be performed vaginally, abdominally—namely the open approach—or laparoscopically or with robot-assisted laparoscopy—namely the minimally invasive approach [41]. Short-term complications include infections, surgical wounds, dehiscence, bleeding, bowel or urinary tract injuries, and other generic surgical complications [40]. Longer-term complications are rare and partly depend on the surgical approach or on the indication for surgery and include urinary incontinence or pelvic organ prolapse and, if the ovaries are not removed, early menopause [41,42].

According to the Organization for Economic Cooperation and Development (OECD) report, the prevalence of hysterectomy is decreasing in most geographic areas thanks to the introduction of less invasive treatment alternatives, including effective medical treatments for excessive bleeding or pelvic pain. Therefore, the variations in hysterectomy rates depend in part on the availability of on the awareness of these alternative approaches. Standardized treatment rates are in the range of 250 per 100,000 females, with extensive cross-country variation (above 350 per 100,000 females in Canada and Germany, and less than 200 per 100,000 females is Spain, Portugal and the Czech Republic) [43]. As for in-country variation, the same study reports two- to three-fold variation across geographic units for most countries, while Canada and the Czech Republic stand out with higher levels of variation.

As reported by the OECD Health Care Utilization Statistics, in 2017 in Italy the number of hysterectomies (including both benign and malignant indications) was of 182.5 for 100,000 females, below the OECD members average of approximately 250 per 100,000 females [44]. Please note that the criteria used by the OECD and by the Italian Ministry of Health to compute the rate of hysterectomies are different, and therefore figures are not comparable. While the OECD considers at the denominator all women, the Italian Ministry of Health includes women over 45 years old. In the Italian context, in the same year in Tuscany there were less procedures (215.79 per 100,000 females over 45 years old) delivered than the national average (244.01 per 100,000 females over 45 years old) [10].

The present study aims at characterizing the differences in the rates of benign hysterectomy in Tuscany (Italy) through a retrospective analysis using administrative data in the years 2016–2018. Therefore, Tuscany Region, through the participation to the Inter-Regional Performance Evaluation System, is the funding body of this research.

Since the 1980s, the international literature has extensively reported great variations for this procedure across different countries and within regions [45,46,47,48]. In 2019, Lungu et al. [21] showed the high extent of unwarranted geographical variation for hysterectomy among the health districts of the Tuscan healthcare system, in terms of treatment rates and waiting times, reporting two- to four-fold variation.

Laparoscopic hysterectomy was first introduced in 1988 [15]. Since then, advancements in technology have allowed more surgeons to offer a rising number of minimally-invasive procedures to women [16]. According to the international literature, vaginal hysterectomy is the preferred approach [41,49]. However, the transvaginal approach has significant limits in the surgical treatment of uterine diseases, including the removal of large uteri or the treatment of patients with pelvic adhesions. Minimally-invasive hysterectomy, defined as removal of the uterus through an abdominal access with the use of small incision and a video laparoscopic technique, is therefore the least invasive approach that allows the treatment of most benign uterine disorders [41]. In 2016 a meta-analysis demonstrated that the two different approaches had similar outcomes in terms of complication rates, ALOS, total operating time and blood loss [50]. Although robot-assisted laparoscopy has provided improved visualization with three-dimensional imaging, improved ergonomics and better mechanics, this approach has proved to be more costly due to higher equipment investments, higher operating time and need for additional surgical training [38,51,52].

Given these premises, this paper analyzes the extent of unwarranted geographical variation of treatment rates and of the adoption of minimally invasive procedures for benign hysterectomy in Tuscany. Our objectives are: (i) to measure the extent of geographical variation in terms of treatment rates, waiting times, and adoption of minimally invasive surgery for benign hysterectomy; (ii) to investigate whether there is a relationship between the three elements mentioned above; (iii) to assess the impact of the surgical approach on the provision of benign hysterectomy, in terms of efficiency (by measuring the ALOS) and efficacy (by measuring the post-operative complications).

## 2. Materials and Methods

After regular meetings where the Inter-Regional Performance Evaluation System data was presented to health professionals and policymakers, we identified the heterogeneous use of the minimally invasive technique among providers as a potential determinant of geographical variation for hysterectomy delivery to Tuscan patients.

A retrospective analysis using the individual level health databases used in the present study include: (i) hospital inpatient data where data are coded using the International Classification of Diseases, 9th revision (ICD9-CM), and (ii) regional health registry data.

Data were anonymized by the Regional Health Information System Office that assigned to each patient an encrypted unique identifier. The study was carried out in compliance with the Italian law on privacy, and approval by an Ethics Committee was not required.

The data management and the analyses were run using SAS (Statistical Analysis System) version 9.4.

The cohort of women was identified by selecting all Tuscan inhabitants aged 18 to 100 who had at least one planned hospitalization for benign hysterectomy in any Italian public hospital in the three-year period from 1 January 2016 to 31 December 2018.

The analysis included all patients who received a benign hysterectomy (the ICDM-9CM codes are reported in the Appendix A) discharged from a Tuscan Obstetrics and Gynecology department. The procedures were divided into minimally invasive (DRG codes: 68.31; 68.41) and open (DRG codes: 68.39; 68.49; 68.9). The codes were chosen in accordance with the codes used to compute performance indicators for the Inter-Regional Performance Evaluation System and were validated in two rounds together with a team of gynecology clinicians who use them for internal performance monitoring purposes. Women undergoing vaginal hysterectomy were removed due to the small number of vaginal hysterectomies performed each year (17, 12, and 9 respectively from 2016 to 2018, representing about 1.2% of the caseload). Records with malignant cancer diagnosis were excluded from the analysis. In case a patient received both open and minimally invasive hysterectomy, we have included both records (in our analysis, *n* = 8).

The age-standardized treatment rates refer to the number of procedures delivered to the women of a health district compared to the population of that district, multiplied by 100,000. The average waiting time for each district considers the sum of the number of days waited by each Tuscan inhabitant from when the operation has been scheduled to the date of the planned hospital admission, divided by the number of procedures performed. Both indicators consider the hospitalization services provided for Tuscan inhabitants regardless of whether they received the service in Tuscany or in another Italian region. The adoption of the minimally invasive technique by Tuscan providers was computed as the number of minimally invasive procedures divided by the total number of benign hysterectomies performed each year, multiplied by 100.

We observed the extent of variation among health districts (*n* = 26) for treatment rates and waiting times, and among Tuscan providers who perform at least 10 benign hysterectomies per year (*n* = 25) for the adoption of the minimally invasive approach.

Starting from the variation in care delivery in 2018 and after verifying that the variables were normally distributed, we ran the Pearson correlation test between treatment rates, waiting times and use of the minimally invasive technique. We considered year 2018 because it is the most recent observation year, and because the extent of variation remained constant over the three years from 2016 to 2018. The analysis of the relationship between treatment rates and waiting times is based on the assumption of Riganti et al. (2017) [53], that if supply and demand are correlated, shortening waiting times and reducing the extent of geographical variation would require supply-side interventions. Furthermore, we investigated the association between waiting times and percentage of minimally invasive procedures for Tuscan providers, to analyze whether a greater adoption of minimally invasive surgery is correlated to a greater imbalance between demand and supply, hence longer waiting times. Finally, we analyzed the relationship between the surgical technique and the treatment rates to assess whether the former could be a determinant of the geographical variation observed for the latter.

After running the correlation analysis, we used the framework proposed by Nuti and Vainieri (2012) [54], where treatment rates and waiting times are plotted in a matrix. The matrix is divided into four different quadrants by the median lines. The first quadrant shows high waiting times and low treatment rates, the second one high waiting times and high treatment rate, the third one short waiting times and low rates, and finally the fourth one short times and high rates.

Average length of stay was computed at a hospital level, by distinguishing between the open and minimally invasive technique. For a further analysis of average length of stay, we split providers into two groups: those who perform more than 30% of their caseload by the minimally invasive approach (group A) and those who perform less than 30% (group B). Providers delivering less than 10 minimally invasive procedures per year were excluded.

Complication rates, both for minimally invasive and for open procedures, were computed as the number of complications, both during the same hospitalization or within 30 days from discharge, divided by the number of cases. The ICD9-CM codes for complications are reported in the Appendix A. Additionally, complication rates were analyzed separately for group A and group B.

Finally, we applied the Pearson correlation test between the average length of stay and the % of minimally invasive procedures delivered by each Tuscan hospital.

## 3. Results

The study cohort included 3302 patients for benign hysterectomy (1068 in 2016, 1094 in 2017 and 1140 in 2018) who are residents living in Tuscany (Italy). The percentage of minimally invasive procedures for benign hysterectomy has been gradually increasing (from 30.2% in 2016 to 35.4% in 2018).

By analyzing the benign hysterectomy data, we observed that geographical variation persisted among the districts in Tuscany. In particular, the age-standardized treatment rates of benign hysterectomy for 2018 registered a minimum value at 20.75 and a maximum equal to 72.29, whereas the mean was 38.56 (SD 12.27). The result is aligned with the previous findings and confirms the existence of geographical variation among the districts in Tuscany [21].

We observed variation also for the waiting times, registering the minimum value equal to 43 days, the maximum equal to 146 days and the mean equal to 85.66 (SD 55). Finally, we analyzed the use of minimally invasive surgical technique and observed again variation. Minimally invasive hysterectomy is not performed by all regional hospitals, with percentages of mini-invasive hysterectomies over conventional open procedures ranging from 0% to 77% (in TH A). Moreover, we also observed an increase in minimally invasive procedures over the three-years period, raising from 30.2% to 35.4%.

The correlation analysis proved no significant association (*p*-value = 0.4758) between waiting times and the surgical technique among Tuscan providers. Analogously, also the relationship between treatment rates and waiting times among health districts turned out statistically not significant (*p*-value = 0.1058).

The results of the age-standardized treatment rates and waiting times were plotted in a matrix, obtaining an intuitive estimation of the extent of geographical variation (Figure 1). This offers the visualization of the variation in four quadrants. In the first quadrant are the health districts with the high waiting times and low treatment rates; the second quadrant shows areas with short waiting times and low treatment rates; the third quadrant incudes long waiting times and high treatment rates and finally, the fourth quadrant has low waiting times and high treatment rates.

Figure 1 illustrates age-standardized treatment rates at district level, divided into open (that varies from 31% to 87%) and minimally invasive technique (from 13% to 69%).

From Figure 2 it seems that there might be a greater use of minimally invasive surgery for women living in health districts that have higher treatment rates. To verify this hypothesis, we ran the correlation test (Figure 3) and obtained a positive (r = 0.4098) and statistically significant (*p*-value = 0.0376) association between the two variables.

At this point we shifted the focus from health districts to care providers, as the measures of efficacy and efficiency are to be sought at the provider level. Indeed, while treatment rates and waiting times might depend also on outside-hospital determinants (e.g., how much the surgical procedure is prescribed by gynecologists), complications and average length of stay depend on the care provider only. Therefore, we analyzed the differences between the open and the minimally invasive approach in terms of average length of stay and post-operative complications.

Results showed that in 2018, but the figures are confirmed also for the other two years, the regional average length of stay for minimally invasive procedures was significantly lower than the one of open surgery (2.85 vs. 4.27 days), with consistent differences between providers. From the analysis between group A (more than 30% of the caseload delivered minimally invasive) and group B (less than 30%) we found out shorter average length of stay for group A with respect to group B. Moreover, the difference holds true both for minimally invasive procedures (3.04 vs. 3.22 days) and for open ones (3.62 vs. 4.89 days). Related graphs are available in the Appendix A.

To investigate deeper the relationship between the percentage of minimally invasive benign hysterectomies and the average LOS, we ran the Pearson correlation analysis. As shown in Figure 4, we observed a negative (coefficient = −0.41) and statistically significant (*p*-value = 0.04) correlation between the two variables.

To deeper analyze the provision of benign hysterectomy from the providers’ viewpoint, we plotted, by former LHAs, the stratification of the providers by surgical technique. We aggregated the health districts (*n* = 26) into LHAs (*n* = 12) with the aim of obtaining comparable observation units, as all health districts do not include in their catchment area a provider where this procedure is performed. We can observe the distribution of benign hysterectomy delivered by the different providers among the LHAs (Figure 5). The histograms represent the treatment rates for the 12 LHAs, a unique color corresponds to each provider and the texture represents the surgical technique (plain for open surgery, hatched for minimally invasive). Among the providers, we also considered those from other regions than Tuscany and we aggregated them into one single provider called “Extra”. A more detailed table is available in the Appendix A.

Finally, investigating the post-operative complications separately for the minimally invasive and open procedures, we observed a difference between the two techniques. Indeed, post-operative complication rate for minimally invasive surgery was 0.5%, while the rate for the open procedures was 2%. We then studied complication rates by distinguishing between providers who more employ the minimally invasive technique (group A) and those who employ it at a lesser extent (group B). Despite complications are a rare event for benign hysterectomy and we analyzed low numbers, we observed differences between the two groups, with providers of group A having lower complication rates than providers in group B. A graphical visualization of the differences is available in the Appendix A.

## 4. Discussion

Benign hysterectomy, as most of the elective surgery procedures, has always been a field of interest for variation scholars since there is no evidence-based standard regarding the optimal treatment rate, and wide differences have been reported in the international literature [47,48]. In terms of the categories identified by Wennberg [19], geographical variation in the provision of benign hysterectomy can be either positive, if depends merely on patients’ preferences, or negative if it is due to supply-side factors. Reducing the extent of avoidable variation is one of the goals of regional and national policymakers, while they should foster the positive one by considering the patients’ perspective. Since variation that must be tackled seems to depend only marginally on patient choices, health services research must consider mainly the perspective of care providers.

Our study confirms the findings already presented by Lungu et al. (2019) [21] that found 2/3-fold variation for benign hysterectomy among the Tuscan districts in 2016. Indeed, the data from 2016 to 2018 show that geographical variation, both in terms of treatment rates and waiting times, persisted over time. The cross-checking of the waiting times and the treatment rates is necessary to point out that if there is a statistically significant and negative relationship between the two variables, it would imply that to reduce waiting times, it would be sufficient to increase the supply. In our case we have observed no significant correlation between the two variables, meaning that the variation of treatment rates does not depend on supply capacity, assuming that need is similar across health districts.

We also found wide differences in the adoption of minimally invasive surgical techniques among providers, ranging from some that employ exclusively the open approach, to hospitals that deliver more than three quarters of hysterectomies by laparoscopy or robot-assisted surgery. This has consequently led to variation when observing how many patients, at a district level, have received the minimally invasive treatment: percentages vary from 13% to 69%. Moreover, we found a positive and significant correlation between the percentage of minimally invasive benign hysterectomies and the standardized treatment rate at a district level. The interpretation of this result must consider that minimally invasive procedures have a shorter average length of stay and therefore allow a quicker turn-over of hospital beds with respect to open procedures. Therefore, the gain in efficiency due to the greater adoption of minimally invasive surgery might translate into an additional capacity to perform procedures. Since the treatment rate in Italy is below the OECD average rate [55], and in Tuscany the treatment rate (215.79 per 100,000 females aged 45 or more) is lower than the national average (244.01 per 100,000 females aged 45 or more), this result has important implications for current practice, because a greater adoption of minimally-invasive surgery could contribute to the reduction of the potential supply gap in some areas and could align the regional treatment rates with the OECD average.

While on the one side minimally-invasive surgery is characterized by greater efficiency (the shorter average length of stay proves it), on the other hand some scholars debate whether more innovative and attractive surgical techniques lead to higher demand from patients and therefore longer waiting lists [56,57]. In our study we observed no statistically significant relationship between the percentage of minimally invasive procedures performed and waiting time.

Additionally, our study confirms the findings of previous investigations that compare minimally invasive and open surgical approaches, showing that the higher the percentage of benign hysterectomies performed, the shorter the average length of stay. Moreover, we observed that providers that employ the minimally invasive technique to a greater extent showed a shorter average length of stay also for open procedures; this finding depicts a potential learning effect that allows providers to shorten length of stay independently on the surgical approach. The results presented so far highlighted the undoubtable benefits of minimally invasive surgery and simultaneously depicted the significant extent of unwarranted geographical variation among Tuscan healthcare providers regarding its adoption.

To discuss the heterogeneous behavior of providers in Tuscany with respect to minimally invasive surgery, we want to focus our attention on the provider that delivers the highest share of minimally invasive benign hysterectomies (77%, THA). There is no doubt that the high percentage of minimally invasive procedures is a proxy of good quality of care, as patients have shorter length of stay and less post-operative complications. As stated in the Results section, despite complications are a rare event for hysterectomies for benign diseases, patients who receive surgery from providers that perform a great share of their caseload by the minimally invasive approach are less likely to have complications within 30 days after discharge. This represents evidence that better outcomes (less postoperative complications) depend mainly on surgical competence, that can only be enhanced by a greater adoption of the minimally invasive technique. Nevertheless, in terms of appropriateness, it is important to understand whether this, together with the high number of procedures delivered per year, leads or not to high treatment rates for the population living in the area (the LHA) where the provider is located. The provider in our example is located within the LHA 5. As we can observe from Figure 5 above, the treatment rate of LHA 5 is close to the regional average (34.58 vs. 37.38 procedures per 100,000 inhabitants), indicating that probably the intensive use of the minimally invasive approach does not lead to high (and inappropriate) treatment rates.

Therefore, it is important to notice that the THA provides the service also for women who come from other LHAs, both from areas with lower treatment rates than the average (LHA 12 and 6) and from areas with higher rates (LHA 2). Therefore, it seems that the model adopted by the THA ensures a great appropriateness of care, as the provision of the service does not imply the risk of over-treatment, and at the same time ensures equity of access to care, as they do not serve only the population in their catchment area.

The systematic use of benchmarked performance data, paired with effective data visualization tools, are fundamental to highlight geographical variations and to raise professionals’ awareness, that leads to a “reputational competition” and in turn contributes to promote change, and hopefully improvement [58]. Moreover, performance measurement and evaluation should be complemented with in-depth analyses to audit practices, through quantitative-based models based on administrative data and through other qualitative or narrative tools based on the patient perspective. These analyses should go beyond the boundaries of the single provider and should cross-check the delivery with the population-based perspective [59]. The elements emerged so far lead us to think that to reduce unwarranted variation and to foster high quality of care, there is a need to standardize the provision of benign hysterectomy at a regional level. One key element of the standardization is the process of training and engagement of health professionals, that together with the acquisition of surgical skills, allows health professionals to have a better understanding of evidence (and differences) at the regional level. For this purpose, the systematic sharing of data and the continuous discussion among health professionals demonstrate that it is possible to achieve greater quality of care and better outcomes without increasing cost [59]. This will enable a greater opinion-sharing among them, reduce self-referentiality, and therefore obtain a more homogeneous delivery of benign hysterectomy by the Tuscan providers.

Finally, the introduction of Patient Reported Outcome Measures (PROMs) along with post-operative complications could facilitate to obtain a more comprehensive measure of hysterectomy patients’ wellbeing through their perspective. PROMs consist into longitudinal surveys, administered to patients before and after receiving the treatment, that through validated questionnaires, aim at obtaining the outcomes of care directly from the point of view of the patients. The health gain is estimated by comparing the score of the pre-operative questionnaire with the one(s) of the follow-up questionnaires, that are generally sent 1, 3, 6, and 12 months after the intervention. This will help to have a better understanding of “grey” outcomes that are so relevant in the landscape of elective surgery, and it could also allow to gain insights regarding the debate over the most appropriate treatment rate to pursue. In fact, as there are no international clinical guidelines that establish which is the appropriate rate of benign hysterectomy, the implementation of systematic PROMs could contribute to the definition of evidence-based criteria and to the reduction of geographical avoidable variation. Furthermore, the collection of PROMs could also contribute to homogenize and reduce waiting times through clinical priority assessment criteria, as already implemented in other international examples such as New Zealand [60].

## 5. Conclusions

The results presented in this paper can be used by policy makers to better manage the provision of benign hysterectomy by simultaneously pursuing three different objectives: reducing geographical variation (treatment rates, waiting time, and surgical technique), increasing benefits for patients (hospital stay and outcomes), and ensuring the financial sustainability of the healthcare system. Indeed, our findings are in line with what has been discussed and proved by international health services scholars: aiming at improving the quality of care allows to pursue also the equity and the financial sustainability of healthcare systems [12,61,62,63,64]. Therefore, policymakers and health managers should focus primarily on quality as it positively impacts the other two pillars of public healthcare. The study considers multiple theoretical perspectives and tests them with the case of benign hysterectomy, but the same reasoning can be applied more widely to other elective (surgical or not) care services. Our study comes with a few limitations: first, it does not consider all kinds of hysterectomy, it focuses only on procedures performed for benign gynecological conditions and with abdominally based surgical approach. Then, it analyzes the Tuscan context, and it focuses on one elective surgical procedure only. However, these two limitations represent a further research opportunity since the framework can be extended to other regional contexts and to other elective surgical procedures.

## Figures and Tables

**Figure 1 ijerph-18-06722-f001:**
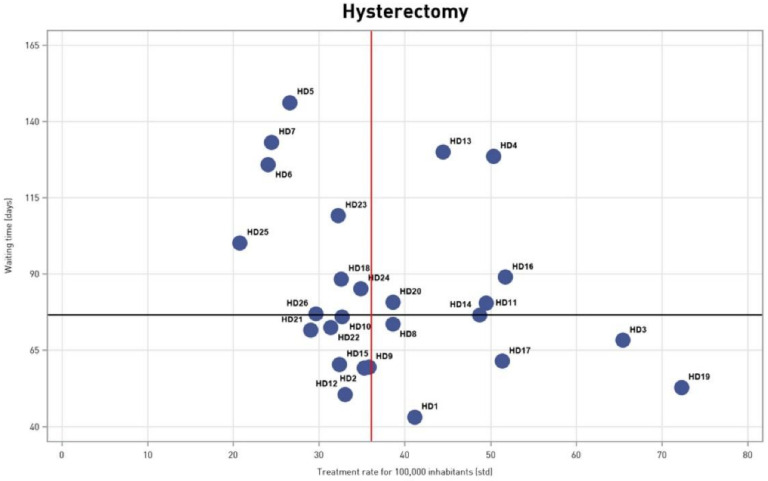
Waiting times and treatment rates for 100,000 inhabitants (std)—2018, Tuscany.

**Figure 2 ijerph-18-06722-f002:**
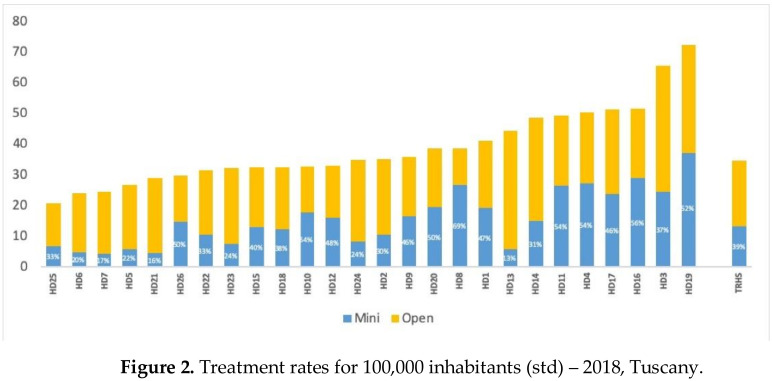
Treatment rates for 100,000 inhabitants (std)—2018, Tuscany.

**Figure 3 ijerph-18-06722-f003:**
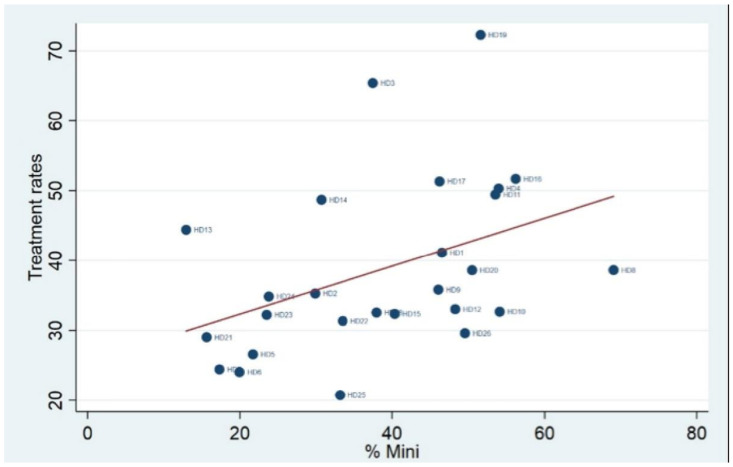
Correlation between treatment rates and percentage of minimally invasive procedures.

**Figure 4 ijerph-18-06722-f004:**
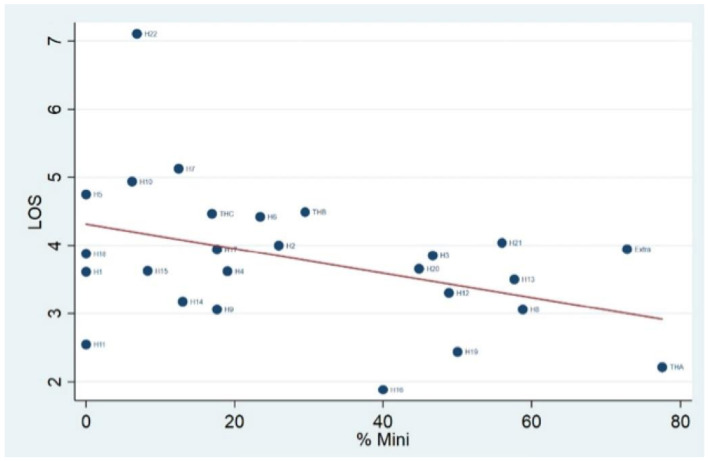
Correlation between average length of stay and use of minimally invasive approach.

**Figure 5 ijerph-18-06722-f005:**
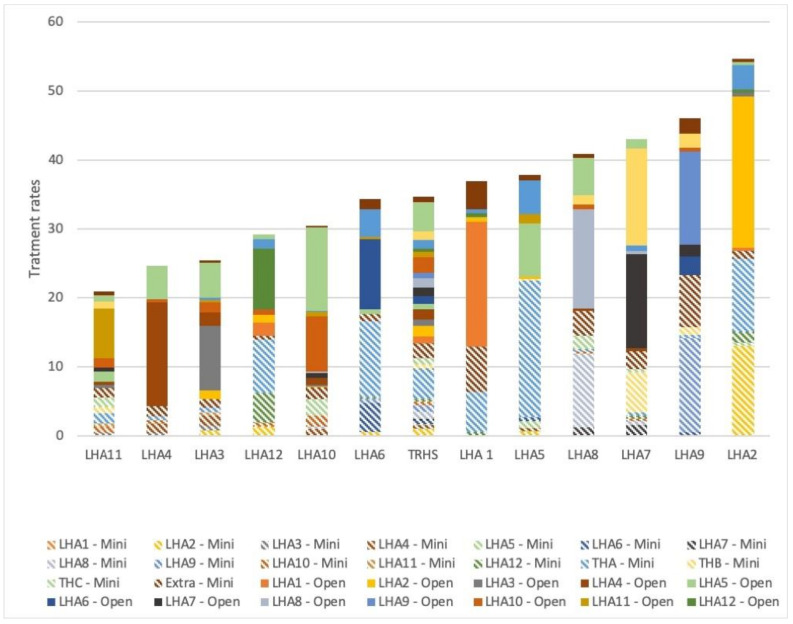
Benign hysterectomy treatment rates for the 12 LHAs, stratified by provider and surgical technique.

## Data Availability

The data presented in this study are available on request from the corresponding author. The data are not publicly available due to the need to request them to the Tuscany Regional Administration Office.

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
