# Peer review of "The Impact of New Surgical Techniques on Geographical Unwarranted Variation: The Case of Benign Hysterectomy"

_ijerph, 2021, doi:10.3390/ijerph18136722_

Round 1
Reviewer 1 Report
In the first part of the report the vaginal hysterectomy is mentioned, but this is not mentioned later in the material and methods or in the results. Why are they not included or please explain the choice to exclude them. What percentage of interventions is vaginal.
Typically, type of surgery (open versus minimal invasive) is determined by the type of pathology. The open approach will lead to longer hospital stay due to increased postoperative pain of the abdominal wall incision. I cannot find any information on the correlation of other outcomes, e.g. weight of the resected uterus or indication for surgery to the length of stay. So how can you compare open procedures to laparoscopic procedures without information on the demographics of the patients involved?
The abbreviation ALOS is not explained at it's first use, please correct
The use of many, I suppose 'selfmade' abreviations (like TH, BH, ES, LOS) does not improve readibility. I would suggest keeping the complete wording instead of the abbreviations, inparticularthe abbreviations mentioned above, as they do not correspond with typically used, widespread abreviations in the field of surgery or gynecology.
I did not find explanations for the abbreviation KPI (page 2)
There's an error in the text on page 8 (after Figure 4)
Author Response
Dear Reviewer,
Many thanks for the revision and for the helpful comments and suggestions. We appreciate your contribution to improving the quality of the manuscript. Please find below our responses to each of your comments and the revised version attached.
- In the first part of the report the vaginal hysterectomy is mentioned, but this is not mentioned later in the material and methods or in the results. Why are they not included or please explain the choice to exclude them. What percentage of interventions is vaginal.
- -> Thank you for your comment. An explanation of the exclusion has now been included at page 5, and it consists in the fact that the percentage of vaginal hysterectomies with respect to the total number of benign hysterectomies is around 1.2%.
- Typically, type of surgery (open versus minimal invasive) is determined by the type of pathology. The open approach will lead to longer hospital stay due to increased postoperative pain of the abdominal wall incision. I cannot find any information on the correlation of other outcomes, e.g. weight of the resected uterus or indication for surgery to the length of stay. So how can you compare open procedures to laparoscopic procedures without information on the demographics of the patients involved?
- -> Thank you for your important comment. We agree that the kind of surgery should depend on the type of pathology, but what we saw in the data is significant variation between hospitals. Indeed, indicators show that different hospitals have massively different adoption of minimally invasive surgery, while the demographics of patients living in Tuscany is to a large extent homogeneous. The main aim of the focus is not to compare the 2 techniques, but to benchmark providers and to investigate if the differences observed in the technique can partly cause the geographical variation observed in the data.
- The abbreviation ALOS is not explained at it's first use, please correct
- -> Thank you for your comment, in line with the next comment, we removed the abbreviation and always kept the extended form throughout the manuscript.
- The use of many, I suppose 'selfmade' abreviations (like TH, BH, ES, LOS) does not improve readibility. I would suggest keeping the complete wording instead of the abbreviations, inparticularthe abbreviations mentioned above, as they do not correspond with typically used, widespread abreviations in the field of surgery or gynecology.
- -> Thank you for the suggestion, we removed the abbreviations throughout the manuscript. The only abbreviation we kept is TH = Teaching Hospital as this is referred in international literature and in our figures 4 and 5. Now the readability is much improved, again many thanks.
- I did not find explanations for the abbreviation KPI (page 2)
- -> Thank you for your comment, we have now replaced the acronym with the extended name “Key performance indicators”.
- There's an error in the text on page 8 (after Figure 4)
- -> Thank you for letting us know, there was a Microsoft Word reference error, we have not removed it.
Reviewer 2 Report
In this study spanning from 2016–2018, the authors analyzed data on benign hysterectomy from Tuscany, Italy aiming to determine whether choice and provision of surgical technique contributed to observed geographic variation in treatment rates and waiting times. Among other results, the authors observed considerable variation among healthcare districts, and in districts where more less invasive procedures were performed, treatment rates were higher.
Although not innovative, this is an interesting paper illustrating variations in procedures and the experience of an Italian region. There are, however, some issues that the authors should address to improve their manuscript:
- Introduction: I believe this section could be shortened considerably, with the authors focusing on the main aims on this study and knowledge from previous research. The amount of information provided now is unnecessarily large with repetitions throughout the whole manuscript.
- Materials and Methods, page 5: Why only variations in 2008 were considered in the analysis?
- Materials and Methods, page 5: Could the authors comment on how reliable (valid) the codes ICD/DRG codes use were to identify the procedures and discriminate among the different techniques?
- Materials and Methods, pages 5–6: Could the authors motivate why Pearson correlation coefficients were estimated for these analyses? Were data normally distributed?
- Results: Codes for the districts and providers shown in the plots should be provided in the footnotes.
- Results: Districts HD13 and 19 appear to be outliers in the analysis of variation in treatment rates in relation to penetration of minimally invasive surgical procedures for benign hysterectomy (presented in Figure 3). Would exclusion of those influence the results in any way?
- Figure 5: This figure is almost impossible to read with so many colors and patterns. I wonder is there is another way to display those results, possibly in a table.
Author Response
Dear Reviewer,
Many thanks for your comments and suggestions. We highly appreciate your contribution to improving the quality of the manuscript. Please find below the reply to your comments and a revised version of the manuscript attached.
In this study spanning from 2016–2018, the authors analyzed data on benign hysterectomy from Tuscany, Italy aiming to determine whether choice and provision of surgical technique contributed to observed geographic variation in treatment rates and waiting times. Among other results, the authors observed considerable variation among healthcare districts, and in districts where more less invasive procedures were performed, treatment rates were higher.
Although not innovative, this is an interesting paper illustrating variations in procedures and the experience of an Italian region. There are, however, some issues that the authors should address to improve their manuscript:
- Introduction: I believe this section could be shortened considerably, with the authors focusing on the main aims on this study and knowledge from previous research. The amount of information provided now is unnecessarily large with repetitions throughout the whole manuscript.
- -> Thank you for the suggestion of shortening the Introduction. We removed unnecessary information and repeated information, and now the manuscript’s flow has significantly improved.
- Materials and Methods, page 5: Why only variations in 2008 were considered in the analysis?
- -> Thank you for your comment, we believe you intended 2018. We started from 2018 as it was the most recent year observed and because the amount of variation remained constant over the three years. However, we included an explanation within the manuscript, at the top of page 6 now.
- Materials and Methods, page 5: Could the authors comment on how reliable (valid) the codes ICD/DRG codes use were to identify the procedures and discriminate among the different techniques?
- -> Thank you very much for your suggestion, this improved the quality of our manuscript. A comment on the reliability and validity of the ICD codes was added at page 5.
- Materials and Methods, pages 5–6: Could the authors motivate why Pearson correlation coefficients were estimated for these analyses? Were data normally distributed?
- -> Thank you for your comment. Yes, we have first verified that the data was normally distributed, we included an explanation in the manuscript (page 6).
- Results: Codes for the districts and providers shown in the plots should be provided in the footnotes.
- -> Thank you for your comment. As this data is not public yet, the Health Regional Administration of Tuscany did not give permission to display the names of districts and providers. This is the reason why only codes are displayed in all our graphs.
- Results: Districts HD13 and 19 appear to be outliers in the analysis of variation in treatment rates in relation to penetration of minimally invasive surgical procedures for benign hysterectomy (presented in Figure 3). Would exclusion of those influence the results in any way?
- -> Thank you for your observation, we must say that we had the same inquiry when we first observed the data. However, we produced the analysis in Figure 3 also after removing HD19, HD13, and also HD3 and results do not change. Therefore we included the complete analysis in the manuscript.
- Figure 5: This figure is almost impossible to read with so many colors and patterns. I wonder is there is another way to display those results, possibly in a table.
- -> Thank you for the suggestion. Indeed the many colours and patters do not help readability. Following your suggestion, we included a table containing the figures in Appendix 4.